# Large-Scale Functional Assessment of Genes Involved in Rare Diseases with Intellectual Disabilities Unravels Unique Developmental and Behaviour Profiles in Mouse Models

**DOI:** 10.3390/biomedicines10123148

**Published:** 2022-12-06

**Authors:** Hamid Meziane, Marie-Christine Birling, Olivia Wendling, Sophie Leblanc, Aline Dubos, Mohammed Selloum, Guillaume Pavlovic, Tania Sorg, Vera M. Kalscheuer, Pierre Billuart, Frédéric Laumonnier, Jamel Chelly, Hans van Bokhoven, Yann Herault

**Affiliations:** 1Université de Strasbourg, CNRS, INSERM, Institut Clinique de la Souris (ICS), PHENOMIN, CELPHEDIA, 1 rue Laurent Fries, 67404 Illkirch, France; 2Université de Strasbourg, CNRS, INSERM, Institut de Génétique et de Biologie Moléculaire et Cellulaire, 1 rue Laurent Fries, 67404 Illkirch, France; 3Max Planck Institute for Molecular Genetics, Research Group Development and Disease, Ihnestr. 63-73, 14195 Berlin, Germany; 4Institute of Psychiatry and Neuroscience of Paris (IPNP), Université de Paris, INSERM U1266, “Genetic and Development of Cerebral Cortex”, 75014 Paris, France; 5GHU Paris Psychiatrie et Neurosciences, Hôpital Sainte Anne, 75014 Paris, France; 6UMR1253, iBrain, University of Tours, Inserm, 37032 Tours, France; 7Service de Génétique, Centre Hospitalier Régional Universitaire, 37044 Tours, France; 8Department of Cognitive Neuroscience, Radboudumc, 6500 HB Nijmegen, The Netherlands; 9Department of Human Genetics, Radboudumc, 6500 HB Nijmegen, The Netherlands; 10Donders Institute for Brain, Cognition, and Behaviour, Centre for Neuroscience, 6525 AJ Nijmegen, The Netherlands

**Keywords:** mouse model, genetic modification, intellectual disability, behavior phenotypes

## Abstract

Major progress has been made over the last decade in identifying novel genes involved in neurodevelopmental disorders, although the task of elucidating their corresponding molecular and pathophysiological mechanisms, which are an essential prerequisite for developing therapies, has fallen far behind. We selected 45 genes for intellectual disabilities to generate and characterize mouse models. Thirty-nine of them were based on the frequency of pathogenic variants in patients and literature reports, with several corresponding to de novo variants, and six other candidate genes. We used an extensive screen covering the development and adult stages, focusing specifically on behaviour and cognition to assess a wide range of functions and their pathologies, ranging from basic neurological reflexes to cognitive abilities. A heatmap of behaviour phenotypes was established, together with the results of selected mutants. Overall, three main classes of mutant lines were identified based on activity phenotypes, with which other motor or cognitive deficits were associated. These data showed the heterogeneity of phenotypes between mutation types, recapitulating several human features, and emphasizing the importance of such systematic approaches for both deciphering genetic etiological causes of ID and autism spectrum disorders, and for building appropriate therapeutic strategies.

## 1. Introduction

Intellectual disability (ID) is a major medical and socio-economic problem owing to its high incidence in the general population. Mutations in about 1500 different genes have been associated with ID [1,2], while pathogenic mechanisms and the molecular basis of gene dysfunction in ID remains to be elucidated. So far, functional studies have mainly focused on single gene defects, such as the Fragile X syndrome. Here, we propose a systematic approach to gain pathway-based insights into mechanisms leading to cognitive dysfunction in humans.

Based on human genetic studies, we selected 45 genes (Table 1) to generate and characterize mutant mice models, of which 39 were selected based on the frequency of pathogenic variants in ID patients. The proteins encoded by these genes act in various biological processes, such as transcription regulation, epigenetic modification, synaptic transmission, or influencing the excitatory/inhibitory balance of central nervous system (CNS) activity. In patients with ID, 31 genes of this selection carry loss-of-function (LoF) variants or deletions, with two genes displaying additional gain-of-function (Gof) mutations, two genes carrying splicing variants, and 12 genes carrying missense variants. Most of these pathogenic variants cause a syndromic ID disorder. Six other genes potentially involved in ID were included, namely *ASCC3* found mutated in a single family by performing a large-scale genomic study [3], *EHMT2* homologous to *EHMT1* causing Kleefstra syndrome, and *CDK8. CDK8* encodes a key subunit of the Mediator complex involved in transcriptional processes and harbouring pathogenic variants in several other subunits, causing syndromic or non-syndromic ID. Two additionally included genes were found mutated in single patients with ID (*CDC27, mIR124a-2*). We also added *STARD8*, which is deleted or disrupted in patients with a contiguous gene deletion and craniofrontonasal syndrome [4,5], and encodes a Rho GTPase-activating protein, like *OPHN1* whose LoF causes X-linked ID [6]. Thirty of the selected genes are located on an autosome and 15 on the X chromosome. Thus, we generated, characterized and compared male mouse models carrying either a mutation identical to a patient with ID or a LoF allele to better understand the function of a candidate ID gene. 

Different types of strategies were followed for generating the appropriate models (see Section 2 and Appendix A). Subsequently, we assessed the viability of the mutant mouse lines. Twenty-seven lines were further investigated using a standardized behavioural screen, focusing on males since a wide range of disorders associated with ID are X-linked ID and 33% of genes in the present study are located on the X-chromosome. Several tests were used to assess a wide range of functions or their pathologies, including circadian activity, neurological reflexes and specific motor abilities, anxiety-related behaviour, sensorimotor gating, and learning and memory processes. For some mouse lines, additional tests were performed to further characterize abnormalities observed or to extend phenotypic traits related to individual genes.

## 2. Materials and Methods

The procedures carried out in this project were performed in agreement with the EC directive 2010/63/UE86/609/CEE, submitted to the French Ethics Committee 017 (Com’Eth) and received accreditation under number 2012-139.

### 2.1. Generation of ID Mouse Models

Different strategies were used to obtain the appropriate mouse lines for genes involved in ID (Appendix A: Mouse models generated and phenotyped in the Gencodys consortium). Four mouse lines were repatriated to serve as reference lines and to extend phenotyping data reported in the literature (*Ehmt1^tm1Yshk^* [72,73], *Il1rapl1^tm1.1Hesk^* [74], *Mecp2^tm1Hzo^* [75]) or to study a candidate ID gene (*Ehmt2^Gt(ES62)Feil^* [76,77]). Forty-six mouse lines were generated as part of the European Conditional Mouse Mutagenesis Program (EUCOMM, 13 lines), Knock-Out Mouse Project (KOMP, 2 lines), the Mouse Genome Project at the WTSI (1 line) or directly in this effort (30 lines); they were all linked to the International Mouse Phenotyping Consortium (IMPC, mousephenotype.org). Mutations were either full knock-out (KO) or KO with conditional potential (40 lines), point mutations (PM, 9 lines) for *Cdc27(S92F), Dnmt3b(D803G), Dync1h1(K3334N), Kif5c(E237V), Med12(R961W), Med17(L369P), Med23(R617Q), Tubb3(M388V), Tubg1(Y92C)*, or humanisation, leading to a duplication of a poly-alanine stretch for *Arx* [10]. All mouse lines were either generated in a C57BL/6N genetic background or backcrossed on the C57BL/6N background. Some mutations induced lethality at the heterozygous or homozygous stage (Figure 1A; Appendix A: mouse models generated and phenotyped in the Gencodys consortium). Therefore, conditional mutants were generated for a selection of genes to overcome this problem, and different alleles were customized to study the potential contribution of specific neuronal (glutamatergic, GABAergic) or non-neuronal (glial) cellular networks using transgenic lines expressing Cre under the control of the *Camk2*a [78], *Dlx5-6* [79] or *Cspg4* [80] promoter.

### 2.2. Embryonic Pipeline 

The viability/sub-viability of mutants was assessed by crossing heterozygous mice and scoring the offspring’s genotypic distribution. Selected lines with homozygous/hemizygous (or heterozygotes) scores below the Mendelian ratio were analysed using the well-defined embryo pipeline that we developed in the IMPC [81]. Briefly, embryos were collected at specific developmental stages to determine their window of lethality (Figure 1B) and further characterize their developmental defects.

### 2.3. Behavioural Phenotyping Strategy

For each mutant line, cohorts of wildtype (WT) and mutant male mice were generated for phenotyping (8–12 mice per genotype). Mice were group-housed (2–4 per cage) and allowed 1–2 weeks acclimation in the phenotyping area with controlled temperature (21–22 °C) under a 12-12 light-dark cycle (light on at 07 a.m.), with food and water available ad libitum. Behavioural testing was performed in 10 to 13-week-old adults and carried out in agreement with EC directive 2010/63/UE86/609/CEE, and under the ethics committee accreditation number 2012-139.

Animals underwent a standardized behavioural screen composed of tests to assess sensory and motor abilities, biological rhythm, pain sensitivity, anxiety-related behaviour, sensorimotor gating, learning and memory, social behaviour, and susceptibility to seizures according to the ARRIVES guidelines [82]. Behavioural protocols are described in the Appendix A; most of them are thoroughly detailed in a recent volume of current protocols [83]. The order of tests was carefully defined from the least to the most stressful test to reduce the potential influence of repeated testing (Figure 1C).

### 2.4. Statistical Analysis

For each mutant line, phenotyping data were analysed using unpaired Student *t*-tests or repeated measures analyses of variance (ANOVA) with one between factor (genotype) and one within factor (time, etc.). Qualitative parameters (e.g., clinical observations) were analysed using the χ2 test. The level of significance was set at *p* < 0.05. A gene/phenotype heat-map was drawn based on *p*-values for each parameter (Appendix A: Gene/phenotype heatmap).

In addition, other analyses were performed on all mutant lines related to categorized biological functions identified using appropriate parameters selected from the different behavioural tests. Fifty-seven (57) parameters were distributed in 10 categories representing biological functions (Appendix A: categories of biological functions and related parameters). For each mutant line, a phenotype score (corresponding to the ratio of number of parameters having a phenotype) was calculated per biological function (Appendix A: the phenotype scores). We considered that a parameter presents a phenotype if its adjusted *p*-value (using Benjamini-Hochberg method to control the false discovery rate within each biological function) was less than 0.05 (Appendix A). Based on these phenotypic scores, Principal Component Analysis (PCA) was performed and a heatmap with cluster representation drawn (see the Section 3).

## 3. Results

### 3.1. Gene/Phenotype Relationship

Fifty mutant lines were generated for 45 genes (Table 1 for genes and abbreviations). The viability of homozygous/hemizygous mutant mice was assessed by crossing the heterozygous mice. From the 50 mutant lines analysed, 66% displayed a homozygous/hemizygous lethality, 8% were sub-viable with less than the expected Mendelian progeny ratio [81], and 26% were viable (Figure 1A). Concerning the lines carrying a point mutation, seven out of nine presented homozygous lethality, and haploinsufficiency or autosomal dominant lethality was also found for *Med12* and *Tubb3*. Finally, of 16 lines with mutated genes on the X-chromosome (all alleles), seven were hemizygous lethal, eight viable and one showed sub-viability (Figure 1A).

This high rate of lethality/sub-viability could be expected, as these genes were selected based on their specific involvement in neurodevelopmental disorders. Indeed, the overall rate of homozygous lethality assessed in IMPC is approximately one out of three mutant genes [81] but increases dramatically when disease-related or essential genes are mutated [84].

To establish the time window of lethality and further characterize these essential genes, 11 mutant mouse lines were assessed (Figure 1B). Two of these lines displayed postnatal lethality (*Setbp1^−/−^*, *Mbd5^−/−^*), dying between birth and eight weeks, three lines displayed perinatal lethality and died during the viability test at E18.5 (*Tubb3^M388V/+^*, *Kdm5c^−/y^, Med25^−/−^)*, two lines died between E9.5 and E12.5 (*Cdk8^−/−^*, *Larp7^−/y^*), while four lines were lethal before E9.5 (*Cdc27^−/−^*, *Ankrd11^−/−^*, *Tti2^−/−^*, *Dync1h1^−/−^*). Three of the 11 selected lines were subjected to embryonic phenotyping (*Setbp1^−/−^*, *Med25^−/−^* and *Tubb3^M388V/+^*), revealing several morphological abnormalities early during development. *Setbp1^−/−^* mutants reacted to forceps stimuli and breathed normally at E18.5. High Resolution Episcopic Microscopy (HREM) and 3D reconstruction data at E15.5 revealed that *Setbp1^−/−^* mice displayed palatal (Appendix A: cranial and cervical vertebra abnormalities of *Setbp1^−/−^* mice at E15.5 and E18.5, panels B, C) and vertebral skeletal defects (Appendix A, panel I), reduced or absent dorsal root ganglia (DRG) (Appendix A, panels E, I) and abnormal nasopharyngeal opening (Appendix A, panel B). Vertebral fusion was confirmed at E18.5 by skeletal staining with red alizarin and alcian blue (Appendix A, panel G). *Tubb3^M388V/+^* mice at E18.5 were smaller than their WT counterparts (Appendix A: cranial nerve and associated ganglia as well as dorsal root ganglia abnormalities in *Tubb3^M388V/M388V^* mutant embryos, panel B), displayed weak or no reaction to forceps stimuli and died between 10 min and 30 min after delivery. HREM data at E15.5 revealed severe reduction of DRG and sensitive nerves in *Tubb3^M388V/+^* mutants (Appendix A, panels E, F), while motor fibres were still present. At the level of cranial nerves, *Tubb3^M388V/+^* mutants displayed agenesis of the trigeminal ganglion (Appendix A, panels H, J), hypoplasia of trigeminal nerves, facioacoustic ganglion and nerves, facial ganglion and nerves, glossopharyngeal and vagus nerves and hypoglossal nerves (Appendix A, panels I–K). Gross morphological analysis of *Med25^−/−^* mutants at E12.5 and E15.5 revealed different ranges of abnormalities including anophthalmia (Appendix A: severe craniofacial abnormalities in *Med25^−/−^* mutant embryos and foetuses, panel B), hypoplasia of the telencephalon (three out of twelve mice) (Appendix A, panel D), rhinocephaly with cyclopia (one out of twelve mice) (Appendix A, panels K, H), proboscis (one out of twelve mice) (Appendix A, panel G), facial clefts (two out of twelve mice) (Appendix A, panel E) and exencephaly (three out of twelve mice) (Appendix A, panels G, J). 

Among the sub-viable mutant lines, *Med17^−/−^* showed decreased body weight, breathing difficulties, and died between P0 and eight weeks after birth. Necropsy examination revealed abdominal and pulmonary haemorrhage, hypoplasia of the thymus and heart failure.

These data strongly support the fact that around 65% of genes involved in ID are essential for survival at normal Mendelian ratios. To better understand the specific involvement of these genes in CNS functions, we generated brain specific conditional mutants. Assessing the viability of mutant lines with a tissue specific *Camk2a* reporter revealed that five out of six lines were viable and one out of six lines was sub-viable.

We pursued the analysis with the standardized behavioural phenotyping of 27 mutant lines to detect a wide range of phenotypic traits affecting different CNS functions. As eight mutant lines were generated for X-linked genes with features found only in males, we focused the adult study on male mutant mice. The first observation of the gene/phenotype heatmap revealed three main classes of mutants based on activity phenotypes observed in the different behavioural tests (Appendix A). The first group of mutated genes inducing a substantial increase in spontaneous activity, the second group of mutant lines with decreased spontaneous activity, and the third group of mutant lines with no change.

#### 3.1.1. Hyperactivity Group

The hyperactivity group includes *Cdk8^Camk2a/Camk2a^* (hereafter named *Cdk8^Camk2a^*), *Ankrd11^Camk2a/Camk2a^* (named hereafter *Ankrd11^Camk2a^*), *Atp6ap2^Camk2a/y^*, *Il1rapl1^−/y^*, *Prps1^Camk2a/y^*, *Ptchd1^−/y^*, *Arx^Dup24/y^*, and *Ascc3^Camk2a/Camk2a^* (hereafter named *Ascc3^Camk2a^*). These eight mutant lines showed a substantial increase in locomotor activity and stereotypic behaviour in different situations including actimetric cages (increased number of beam-breaks), the open field (higher distance travelled over the 30 min test) (Figure 2), the Y-maze, and social tests (increased number of visits) [85,86]. They also showed other behavioural alterations. For example, all *Atp6ap2^Camk2a/y^* [85], *Il1rapl1^−/y^* and *Ptchd1^−/y^* [86] mutants had altered contextual and cued fear conditioning, displaying reduced percentage of freezing both during the context and the cued testing sessions (Figure 2). *Ankrd11^Camk2a^* also had decreased contextual fear conditioning. In addition, *Il1rapl1^−/y^* mice had altered spatial learning in the water maze with reduced number of platform crosses, while *Ptchd1^−/y^* had decreased working memory both in the Y-maze and the object recognition tasks [86] (Figure 2). They also showed altered motor abilities, evidenced by decreased muscle strength in *Ptchd1^−/y^* and *Cdk8^Camk2a^* mutants (Figure 2) and decreased startle response in the *Atp6ap2^Camk2a/y^* mice [85].

#### 3.1.2. Hypoactivity Group

Mutant lines constituting this group (*Ehmt1^+/−^, Ehmt1^+/−^/Ehmt2^+/−^, Mbd5^+/−^, Cdkl5^−/y^*, *Mecp2^−/y^*) showed decreased locomotor activity in the actimetric cages, in the social test and in the Y-maze evidenced by reduced number of beam breaks and reduced number of entries/visits (Figure 3). *Ehmt1^+/−^, Ehmt1^+/−^/Ehmt2^+/−^* and *Cdkl5^−/y^* mutants were also less active than their matched wildtypes in a novel environment in the open field, showing lower distance travelled than their corresponding wildtypes (Figure 3).

This group of hypoactive mutants also had other behavioural alterations. *Mbd5^+/−^
*had decreased contextual and cued fear conditioning with decreased freezing performance, improved sociability exploring more the congener than an object, but had altered social memory displaying no preference of novel congener, and decreased startle response (Figure 4). On the other hand, *Ehmt1^+/−^, Ehmt1^+/−^/Ehmt2^+/−^
*had altered recognition memory with recognition index around the chance level, altered social memory (for *Ehmt1^+/−^*) and increased startle response (Figure 4). *Cdkl5^−/y^* displayed decreased prepulse inhibition (PPI) and increased thermal pain threshold (Figure 4), while *Mecp2^−/y^* mutants had decreased startle reactivity and PPI, showed substantial tremors (100% of accuracy), and had a decreased thermal pain threshold. Finally, all these hypoactive lines, except *Mecp2^−/y^*, which showed the opposite phenotype, displayed a trend of increased anxiety with either decreased exploration of the centre of the open field during the first 5 min and decreased object exploration (*Ehmt1^+/−^, Ehmt1^+/−^/Ehmt2^+/−^*) or increased latencies to exit the start arm in the Y-maze (*Mbd5^+/−^, Cdkl5^−/y^*) (Figure 4). *Mecp2^−/y^* mutants instead showed decreased marble burying, increased activity in the open field during the first 5 min, and increased open arm exploration in the elevated plus maze, reflected by increased head dips, tendency to increased percentage of time and decreased entry latency in the open arms.

#### 3.1.3. No Activity Phenotype Group

Finally, a group of mutant lines including *Nr1i3^−/−^*, *Dyrk1a^Camk2a^*, *Dyrk1a^Dlx5,6/+^*, *Stard8^−/y^*, *Wdr62^−/−^*, *Kif5c^+/−^*, *miR137^+/−^*, and *Ehmt2^+/−^* did not show any strong pattern of modified activity and displayed only a limited number of behavioural changes.

### 3.2. PCA and Cluster Analysis

Based on association studies, additional statistical analysis was performed on the 21 mutant lines whose genes are closely associated with ID (six genes were excluded). A graphical representation of phenotype scores was done using a heatmap combined with a dendrogram showing the arrangement of mutant line clusters produced by hierarchical clustering (Figure 5 and Appendix A). Correlated variables were grouped together on a circle of correlations. The smaller an individual coordinate on an axis, the smaller its contribution to the component. The three first components explained 70.27% of the variance (Figure 5A).

The first observation of Figure 5B shows a group of mutant lines, including *Atp6ap2^Camk2a/y^*, *Il1rapl1^−/y^*, *Ehmt1^+/−^*, *Mbd5^+/−^, Cdkl5^−/y^*, and *Ptchd1^−/y^*, with a high number of functional alterations. The second cluster includes gene mutations with a moderate number of altered functions, and includes *Mecp2^−/y^*, *Setbp1^+/−^*, *Entpd1^−/−^*, *miR137^+/−^*, *Dyrk1a^Dlx5−6/+^*, *Wdr62^−/−^*, *Prps1^Camk2a/y^*, *Cdk8^Camk2a^*, *Ankrd11^Camk2a^*, *Arx^Dup24/y^*, and *Cntnap2^−/−^*. The last group of mutants displayed a few changes or no phenotype. PCA was performed to visualise potential links between biological functions and mutant line similarities (Figure 5C,D and Appendix A). On the one hand, a group of lines including *Cdkl5^−/y^*, *Il1rapl1^−/y^*, *Ptchd1^−/y^*, *Atp6ap2^Camk2a/y^*, *Mbd5^+/−^*, and *Ehmt1^+/−^* displayed alterations mainly in activity, repetitive behaviour, novelty exploration and anxiety (Axis 1). On the other hand, mutant lines including *Cdk8^Camk2a^*, *Mecp2^−/y^* and *Setbp1^+/−^* showed deficits in motor abilities and pain sensitivity, while *Atp6ap2^Camk2a/y^* and *Ilrapl1^−/y^* showed learning and memory deficits.

## 4. Discussion

In the present study, we generated 50 mutant lines for 45 genes clinically or potentially relevant for further understanding ID in humans. About 66% of the mutant lines generated were homozygous/hemizygous lethal, providing evidence that these genes are essential for normal development and survival. Embryonic phenotyping of homozygous/hemizygous/heterozygous lethal lines revealed several abnormalities, including pronounced craniofacial and skeletal defects, severe ganglia hypoplasia, and abnormal nervous or sensory system development in line with congenital malformations observed in the corresponding human syndromes. For example, we found that *Setbp1^−/−^* mutants displayed palatal and vertebral skeletal defects, reduced DRG and abnormal nasopharyngeal opening, reproducing some aspects of the SETBP1 Disorders (also known as Mental Retardation, Autosomal Dominant 29), characterized by ID and distinctive facial features [61,87]. We also found in adult *Setbp1^+/−^* mice several behavioural alterations including muscle weakness and altered PPI reminiscent of symptoms observed in patients with a similar mutation type [88]. This model is of interest for better understanding the physiopathology of this new syndrome. Our results from *Med25^−/−^* embryos revealed several abnormalities including exencephaly, anophthalmia or cyclopia and telencephalon hypoplasia, in line with those observed in humans with *MED25* mutations such as Basel-Vanagaite-Smirin-Yosef syndrome characterized by severely delayed psychomotor development resulting in ID, as well as variable eye, brain, cardiac, and palatal abnormalities [47]. Our results obtained for *Tubb3^M388V/+^* embryos are in line with those previously reported for the *Tubb3^R262C/R262C^
*mouse model and with a spectrum of abnormalities including hypoplasia of oculomotor nerves and dysgenesis of the corpus callosum and anterior commissure observed in human syndromes with TUBB3 mutations [66,67,89], supporting the role of TUBB3 in axonal guidance and maintenance.

Large scale standardized behavioural phenotyping of 27 mutant lines carrying mutations in genes involved in ID in humans revealed unique gene/phenotype behavioural profiles based on activity patterns. Interestingly, genes mutated in the hypoactive group are altered either in Kleefstra syndrome (*Ehmt1, Mbd5, Nr1i3*) or Rett syndrome (*Mecp2*) or CDKL5 Deficiency Disorder, CDD (*Cdkl5)*. Kleefstra syndrome is characterized by ID, childhood hypotonia, severe expressive speech delay and a distinctive facial appearance with a spectrum of additional clinical features including autistic-like behavioural problems and cardiac defects [25,26,50,90]. Autosomal Dominant Mental Retardation 1/2q23.1 deletion syndrome, caused by pathogenic *MBD5* variants, shares several phenotypic traits with Kleefstra syndrome [36,37]. Similarly, patients with a pathogenic variant in *CDKL5* (CDKL5 Deficiency Disorder, CDD) or *MECP2* (Rett syndrome) present with overlapping clinical features.

We report here that *Ehmt1^+/−^, Ehmt1^+/−^/Ehmt2^+/−^
*and *Mbd5^+/−^* mice show hypoactivity and learning and memory deficits in several situations, extending previously reported data in *Ehmt1^+/−^
*mice [72,91], and recapitulating several cognitive, autistic and hypoactivity features observed in Kleefstra syndrome and 2q23.1 deletion patients, supporting the use of *Ehmt1^+/−^* and *Mbd5^+/−^* as good mammalian models for Kleefstra and Autosomal Dominant Mental Retardation 1/2q23.1 deletion syndromes.

We found only limited and weak behavioural changes in *Ehmt2^+/−^* and *Nr1i3^−/−^* mutants. *EHMT2* and its paralog *EHMT1* encodes a histone methyltransferase and act together in protein complexes responsible for deposition of mono- and di-methylated forms of Histone 3 Lysine 9 (H3K9me/me2). These methylation marks are associated with gene silencing in euchromatin [92]. Since LoF variants in *EHMT1* give rise to Kleefstra syndrome, it is tempting to speculate that *EHMT2* is a candidate for syndromic ID as well. However, our data from *Ehmt1^+/−^, Ehm2^+/−^* and *Ehmt1^+/−^/Ehmt2^+/−^* mutants show that the main phenotypic traits are linked to the *Ehmt1^+/−^* mutation, in line with a previous study where we showed that unlike *Ehmt1^+/−^, Ehmt2^+/−^* did not present the marked increase of H3K9me2/3 [76] reduces the strength of the hypothesis linking *EHMT2* with Kleefstra spectrum disorders and is potentially associated ID. Indeed, several *EHMT2* LoF alleles have been reported without convincing evidence for involvement in human genetic disorders. For *NRI3*, a single de novo missense mutation (c.740T>C [p.Phe247Ser]) was identified in a patient with core symptoms of Kleefstra syndrome [50]. In our study, *Nr1i3^−/−^* mice displayed only a slight decrease in contextual fear conditioning. In line with behavioural data, the assessment of hippocampal neuronal morphology in *Nr1i3^−/−^* mice did not reveal any gross abnormality concerning neurite length, branching or excitatory synapse density (not shown). Elsewhere, the quantification of ectopic wing vein formation in Drosophila [50] revealed that the EHMT overexpression phenotype was almost completely rescued by heterozygous LoF mutations in *EcR/Nr1i3*, and overexpression of *EcR/Nr1i3* enhanced EHMT-induced ectopic vein formation, providing strong evidence of a synergistic relationship between EHMT and EcR/NR1I3, and that *NR1I3* per se has reduced incidence. Combined, the results could suggest that in the patient affected, the (c.740T>C [p.Phe247Ser]) single amino acid substitution is not a loss of function mutation and has a different effect on the protein. Additionally, the patient carried a de novo *MTMR9* missense variant (c.310T>G [p.(Ser104Ala)]; NM_015458.3) of uncertain significance [50].

Several mutant lines showed a characteristic hyperactivity phenotype. *Il1rapl1^−/y^*, *Ptchd1^−/y^*, and *Arx^Dup24/y^* mice displayed substantial hyperactivity and stereotypic behaviour, and either increased exploration or reduced anxiety. These mutant lines also had altered learning and memory abilities. In humans, *IL1RAPL1* and *PTCHD1* mutations are found in X-linked ID or X-linked autism spectrum disorders [32,58,59]. Among behavioural features of these syndromes, hyperactivity, stereotypies, and altered learning abilities are commonly present. Interestingly, motor problems including psychomotor delay and hypotonia present in patients with *PTCHD1* or *ARX* mutations were also found in our mutants, which displayed either decreased muscle strength (*Ptchd1^−/y^*) or altered grasping and reaching, reflecting fine-tuned motor abilities (*Arx^Dup24/y^*) [9,10,93]. In human syndromes, mutations are hemizygous substitutions or deletions (*IL1RAPL1*), hemizygous deletion or insertion (*PTCHD1*), or hemizygous c.428_451dup24 duplication (ARX). Our mutant lines reproduced some of these mutation types.

Mutations in *ANKRD11*, *ATP6AP2* and *PRPS1* have also been associated with several neurodevelopmental disorders with ID in humans [7,11,94,95]. *ATP6AP2* mutations are found in X-linked ID with Parkinsonism and spasticity, and *PRPS1* mutations in Arts syndrome, X-linked recessive Charcot-Marie-Tooth disease 5 and X-linked non-syndromic hearing loss [11,55,56,57,95,96,97,98]. Mutations are either hemizygous splice site mutations that leads to a LoF (*ATP6AP2*), or substitutions leading mainly to GoF, but also to LoF (*PRPS1*), causing different behavioural symptoms including hyperactivity, stereotypies and altered cognitive abilities. On the other hand, heterozygous deletions or splice site mutations in the *ANKRD11* gene have been found in patients with KBG syndrome, characterized by macrodontia, distinctive craniofacial and skeletal anomalies, short stature, and neurological problems including ID [7,94]. Hyperactivity, anxiety, and hearing loss have also been described [99]. In the present study, we generated *Atp6ap2^−/y^*, *Prps1^−/y^* and *Ankrd11^−/−^* mutant lines and found them embryonic lethal. We then generated and characterized the neuronal specific lines *Ankrd11^Camk2a^*, *Atp6ap2^Camk2a/y^*, and *Prps1^Camk2a/y^*. *Ankrd11^Camk2a^* and *Atp6ap2^Camk2a/y^* displayed substantial hyperactivity and stereotypic behaviour, increased exploration and reduced anxiety, and altered learning and memory. Interestingly, *Atp6ap2^Camk2a/y^* also showed decreased startle response in line with motor problems including hypotonia found in patients. Our data reproduced some of the behavioural phenotypes observed in patients and support, at least in part, the specific effect of the deletion on excitatory neuronal cells. *Ankrd11^+/−^* mutants also showed hearing loss displaying increased ABR thresholds [100], in line with hearing problems reported in some KBG patients [99]. Among the hyperactive lines, *Prps1^Camk2a/y^* showed only subtle phenotypes displaying increased stereotypic behaviour and increased working memory. We assume that neuronal loss-of-function per se is unlikely to model syndrome-related behavioural traits. In line with this observation, we found that *Prps1^Csp4/y^* mice, a glial specific conditional KO, displayed increased stereotypic behaviour during the initial exploration in the actimetric cages, and decreased motor performance in the rotarod (Appendix A).

In the present study we also analysed the effect of LoF variants in several candidate ID genes. One of these genes, *CDK8*, encodes an important regulator of the multi-subunit Mediator complex, involved in transcriptional processes. Mutations in other subunits of the Mediator complex were previously identified in syndromic and non-syndromic ID. *Cdk8^−/−^* mice are lethal; therefore, we generated and analysed *Cdk8^Camk2a^* mice. These mutants displayed hyperactivity in several situations, a trend to increased stereotypic behaviour, hypotonia reflected by decreased muscle strength, and altered recognition memory. While our studies were ongoing, various heterozygous *CDK8* missense mutations were reported to cause a syndromic developmental disorder characterized by hypotonia, ID, and behavioural abnormalities [13]. Affected individuals tended to have learning disability, autistic features and attention deficit-hyperactivity disorder (ADHD). In vitro functional studies showed that the mutations strongly attenuated CDK8 kinase activity, supporting a dominant-negative mechanism of pathogenesis for *CDK8* substitutions [13]. Interestingly, our data clearly recapitulate most behavioural alterations described in humans [13], and suggest that these alterations are neuronal specific.

*ASCC3* was identified as a candidate gene for autosomal recessive ID, with a potentially pathogenic missense variant (p.(S1564P)) identified in a single family [3]. In the present study, we show that *Ascc3^−/−^* mice are lethal, and *Ascc3^Camk2a^
*mutants showed hyperactive behaviour and increased rearing in the actimetric cages, and reduced anxiety-related behaviour in the elevated plus maze. We suggest that neuronal homozygote deletion of *Ascc3* gene is not sufficient to induce profound behavioural alterations. The family affected by the *ASCC3* variant was also reported to have mild ID [3].

Across this study, applying an extensive behavioural pipeline allowed us to identify different classes of genes for which mutations caused several behavioural alterations. The heterogeneity of phenotypes and penetrance are reminiscent of the effects of mutations observed in human syndromes with ID. The class of genes with increased activity includes several *Camk2a*-conditional KO lines specific to excitatory neurons. It can be suggested that the hyperactive phenotype might be due, at least in part, to the *Camk2a* promoter driving Cre recombinase. Hyperactivity and stereotypies are almost common phenotypes in human syndromes with ID. In addition, our constitutive *Il1rapl1^−/y^* and *Ptchd1^−/y^*, and *Arx^Dup24/y^* mice also showed substantial hyperactivity in different situations. Finally, behavioural phenotyping of the *Camk2a-Cre* reporter line did not show any obvious sign of hyperactivity or another relevant behavioural alteration (not shown). These arguments reduce the strength of the hypothesis that hyperactivity observed in the *Camk2a-mutant* lines is likely related to the *Camk2a* promoter driving Cre recombinase. Nevertheless, it is noteworthy that *Camk2-Cre* specific mutations for constitutively lethal lines are partially expressed (in the glutamatergic neurons). Additional data from mutants bred on other reporter lines would increase our knowledge about the effect of mutations described in this study depending on their expression in other cellular compartments. Conditional mutants with combined expression in different cell types might potentially display extensive or stronger phenotypic traits affecting more functions.

Our functional findings are based on a thorough behavioural exploration of gene mutations related to ID in mice, focussing on males. It should be emphasized that around 33% of genes involved in ID (and those generated here) are X-linked genes. That is the main reason why only males were characterized in this study, although this might also be considered a limitation. The extension of phenotyping to females might have increased the strength of our findings. In this regard, in the context of another worldwide effort designed to generate and characterize null mouse models for all the genes of the mouse genome, we also characterized mutant females with some of the genes from the present collection. For example, *Ptchd1^−/−^* females also displayed several behavioural abnormalities including hyperactivity and cognitive deficits, extending the data observed in *Ptchd1^−/y^* males [86].

## 5. Conclusions

The results of the present study allowed us to establish a broad gene-phenotype relationship map for a wide range of genes involved in several neurodevelopmental disorders with ID, or potentially involved in ID. Several of the mutant lines studied reproduced, as far as possible, the human mutation types, and displayed strong phenotypic similarities with patient features, constituting interesting genetic tools to better understand human syndromes with ID and making it possible to establish new potential therapeutic strategies.

## Figures and Tables

**Figure 1 biomedicines-10-03148-f001:**
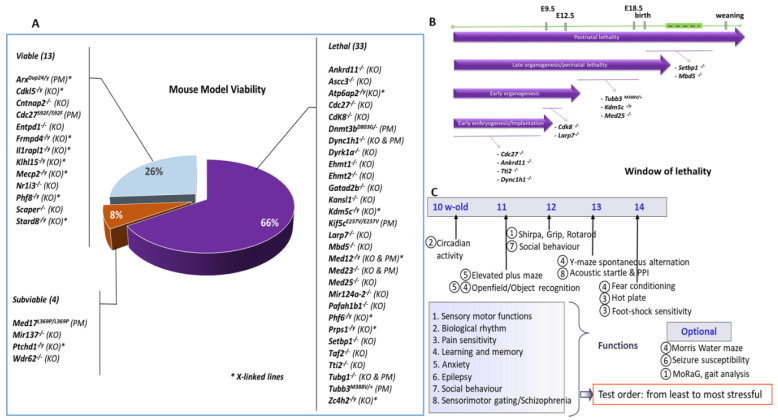
(**A**) Schematic representation of lethality. The percentage of lethal, sub-viable and viable mutant lines is presented with the list of lines for each category. (**B**) Time window of lethality analysed for 11 selected lethal lines. (**C**) Behavioural phenotyping scheme. Several tests were used for assessing a wide range of CNS functions; tests are ordered from least to most stressful.

**Figure 2 biomedicines-10-03148-f002:**
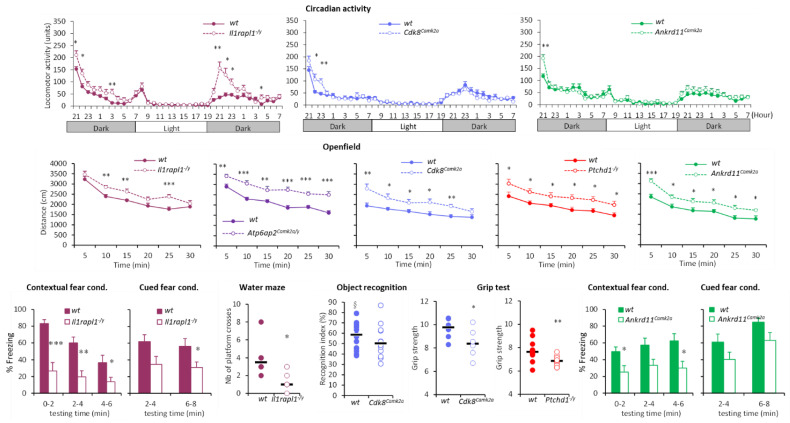
Selected behavioural alterations in the hyperactivity group. *Cdk8^Camk2a^*, *Ankrd11^Camk2a^*, *Atp6ap2^Camk2a/y^*, *Ilrapl1^−/y^* and *Ptchd1^−/y^* [86] showed increased activity in the circadian activity and open field tests reflected by a higher number of beam breaks and distance travelled compared to wild-types. Data are expressed as the mean ±SEM of front and back successive beam breaks (circadian activity) and distance (open field) across time, and analysed using repeated measures ANOVA, followed by *t*-tests for each time point. *Cdk8^Camk2a^*, *Ankrd11^Camk2a^*, and *Ilrapl1^−/y^* showed altered learning performance in the water maze (decreased number of platform crosses for *Ilrapl1^−/y^*), fear conditioning (decreased percentage of freezing for *Ankrd11^Camk2a^* and *Ilrapl1^−/y^*) or object recognition (*Cdk8^Camk2a^*), or motor deficits in the grip test (*Cdk8^Camk2a^* and *Ptchd1^−/y^*). Data are expressed as mean ± SEM% freezing (fear conditioning), or scattergrams with the median for number of platform crosses (water maze), recognition index (object recognition), or muscle strength (grip test), and analysed using either repeated measures ANOVA followed by *t*-tests, or Student *t*-tests for single time points. * *p* < 0.05, ** *p* < 0.01, *** *p* < 0.001 vs. WT; § *p* < 0.05 vs. the chance level (only WT displayed good performance while mutants performed at the hazard level).

**Figure 3 biomedicines-10-03148-f003:**
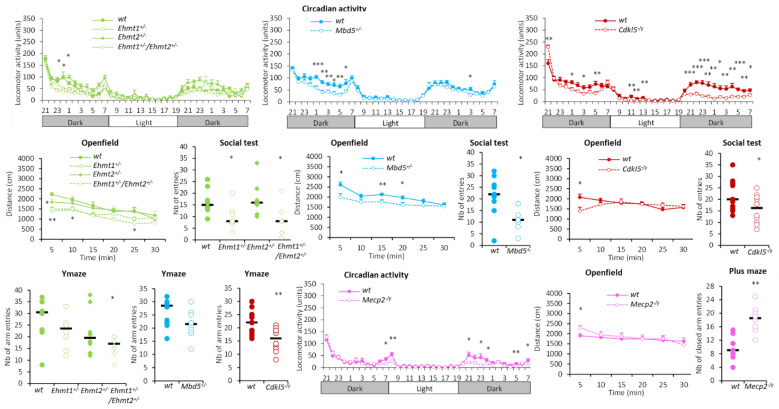
Hypoactivity group. *Ehmt1^+/−^, Ehmt1^+/−^/Ehmt2^+/−^, Mbd5^+/−^* and *Cdkl5^−/y^* showed decreased activity in the circadian activity, in the open field, in social recognition and in the Y-maze tests, reflected by lower number of beam breaks, reduced distance, and lower number of visits or arm entries, respectively, as compared to wild-type counterparts. *Mecp2^−/y^* showed decreased locomotion in circadian activity and increased closed arm entries in the elevated plus maze. Data are expressed as the mean ± SEM of front and back beam breaks (circadian activity) distance (open field) across time, or as scattergrams with the median for the number of entries (social, Y-maze, plus maze tests) and analysed using repeated measures ANOVA, followed by a *t*-test for each time point. * *p* < 0.05, ** *p* < 0.01, *** *p* < 0.001 vs. WT.

**Figure 4 biomedicines-10-03148-f004:**
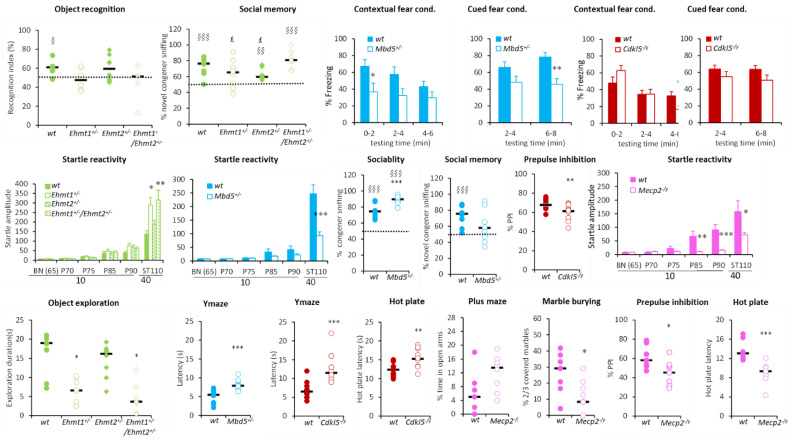
Selected behavioural alterations in the hypoactivity group. *Ehmt1^+/−^, Ehmt1^+/−^/Ehmt2^+/−^, Mbd5^+/−^* showed altered memory performance in the fear conditioning (decreased % of freezing), object recognition (recognition index) or social tests (% novel congener sniffing). *Mbd5^+/−^* and *Mecp2^−/y^* had decreased startle while *Ehmt1^+/−^, Ehmt1^+/−^/Ehmt2^+/−^* had improved startle. *Ehmt1^+/−^, Ehmt1^+/−^/Ehmt2^+/−^, Mbd5^+/−^* and *Cdkl5^−/y^* showed increased anxiety in the open field (decreased object exploration) or in the Y-maze (increased latency). Mecp2 mutants show decreased anxiety in marble burying (decreased number of marbles covered) and in the plus maze (increased %time in the open arms) and decreased pain threshold. Data are mean ± SEM (fear conditioning, startle reactivity) or scattergrams with the median (object recognition, social test, Y-maze, marble burying, hot plate). Data are analysed using either repeated measures ANOVA followed by *t*-tests or Student’s *t*-test for single time points. * *p* < 0.05, ** *p* < 0.01, *** *p* < 0.001 vs. WT; § *p* < 0.05, §§ *p* < 0.01, §§§ *p* < 0.001 vs. the chance level; £ vs. *Ehmt1^+/−^/Ehmt2^+/−^*.

**Figure 5 biomedicines-10-03148-f005:**
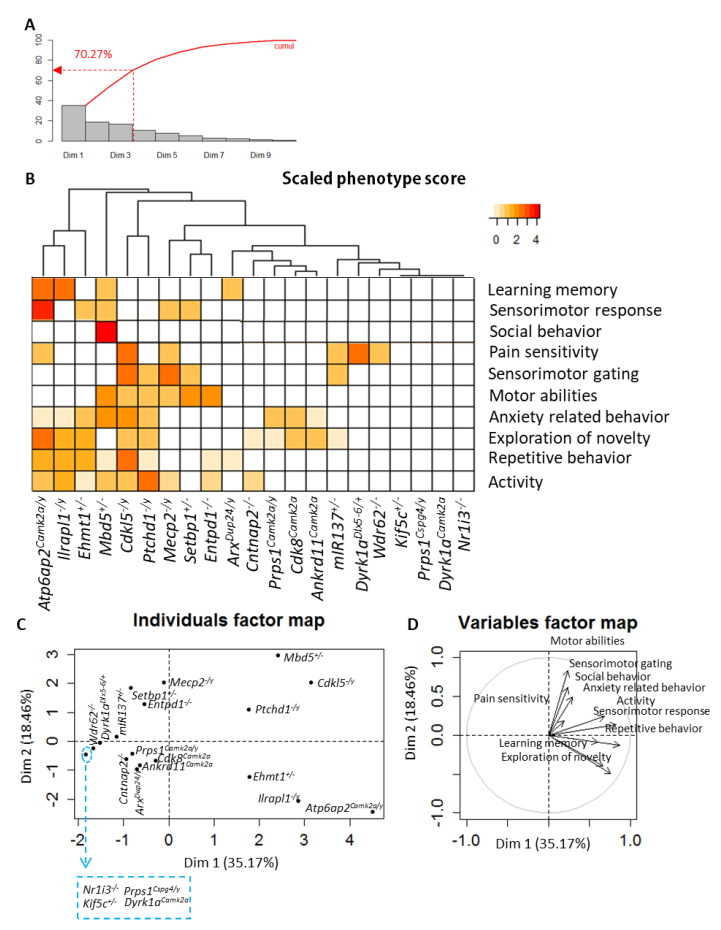
(**A**) PCA—Percentage of explained variance in principal component analysis for each component and cumulated percentage of explained variance. (**B**) Heatmap of scaled phenotype scores data. Row dendrograms show the distance between mutant lines and arrangement produced by hierarchical clustering. Colours indicate the phenotype score level. The darker the orange, the more phenotype parameters the biological function has. (**C**) PCA—Individuals factor map shows in axes 1 and 2 similarity of components between mutant lines. Note: 4 mutant lines listed in the blue square have the same coordinates. (**D**) PCA—Variables factor maps or correlations circle shows in axes 1 and 2 links between biological functions and the axes.

**Table 1 biomedicines-10-03148-t001:** List of mutated genes, human variants, their functional implication in human syndromes, and mouse models generated in this study for these genes. LoF and GoF indicate loss-of-function and gain-of-function, respectively. SNPs indicate Single Nucleotide Polymorphism.

*Gene*	Chr	Human Variant(s)	Syndrome(s)	References	Mouse Models
*Ankrd11*	8	LoF mutations and deletion (heterozygous)	KBG Syndrome	[7]	*Ankrd11^Camk2a^*
*Arx*	X	LoF mutations, deletion, duplication/expansion (hemizygous) 24bp duplication most frequent mutation	Early infantile epileptic encephalopathy 1, Hydranencephaly with abnormal genitalia, X-linked lissencephaly 2, X-linked Mental retardation, Partington syndrome, Proud syndrome	[8,9,10]	*Arx^Dup24/y^*
*Ascc3*	10	-	-	-	*Ascc3^Camk2a^*
*Atp6ap2*	X	Splice site and missense (hemizygous)	ID +/− Parkinsonism with spasticity	[11,12]	*Atp6ap2^Camk2a/y^*
*Cdc27*	11	-	-	-	*Cdc27^−/−^; Cdc27^S92F^*
*Cdk8*	5	Missense substitutions	-	[13]	*Cdk8^−/−^; Cdk8^Camk2a/y^*
*Cdkl5*	X	Translocations, microdeletions, missense, LoF mutations & mosaic exonic deletions (hemizygous in males & heterozygous in females)	CDKL5 Deficiency disorder (CDD) & Early Infantile Epileptic Encephalopathy 2 (EIEE2)	[14,15]	*Cdkl5^−/y^*
*Cntnap2*	6	LoF mutations & SNPs (homozygous or compound heterozygous)	Cortical Dysplasia-Focal Epilepsy Syndrome (CDFES), Pitt-Hopkins like syndrome 1 & Autism & Specific Language Impairment	[16,17,18]	*Cntnap2^−/−^*
*Dnmt3b*	2	LoF mutations (homozygous or compound heterozygous)	Immunodeficiency-centromeric instability-facial anomalies syndrome 1	[19,20]	*Dnmt3b^D803G/−^*
*Dync1h1*	12	LoF mutations (heterozygous)	Autosomal dominant axonal Charcot-Marie-Tooth type 20 disease (CMT20), Autosomal dominant mental retardation 13 (MRD13), Autosomal dominant lower extremity-predominant Spinal muscular atrophy 1 (SMALED1)	[21,22,23]	*Dync1h1^−/−^; Dync1h1^K3334N^*
*Dyrk1a*	16	Translocations, LoF mutations & deletions (heterozygous)	Autosomal Dominant Mental Retardation 7 (MRD7)	[24]	*Dyrk1a^−/−^; Dyrk1a^Camk2a^; Dyrk1a^Dlx5−6/+^*
*Ehmt1*	2	Translocations, microdeletion & LoF mutations (heterozygous)	Kleefstra syndrome	[25,26]	*Ehmt1^+/−^*
*Ehmt2*	17	-	-	-	*Ehmt2^+/−^*
*Entpd1*	19	LoF mutations (homozygous)	Autosomal Recessive Spastic Paraplegia 64 (SPG64)	[27]	*Entpd1^−/−^*
*Frmpd4*	X	LoF mutations, missense and exon deletion (hemizygous)	X-linked Intellectual disability & Schizophrenia	[28,29]	*Frmpd4^−/y^*
*Gatad2b*	3	LoF mutations & deletions (heterozygous)	Intellectual disability	[30]	*Gatad2b^−/−^*
*Il1rap1l1*	X	LoF mutations (hemizygous)	X-linked intellectual disability	[31,32]	*Il1rpl1^−/y^*
*Kansl1*	11	LoF mutations & deletions (heterozygous)	Koolen-De Vries syndrome	[33]	*Kansl1^−/−^*
*Kdm5c*	X	LoF mutations (hemizygous)	Claes-Jensen type X-linked syndromic mental retardation	[34]	*Kdm5c^−/y^; Kdm5c^Camk2a/y^; Kdm5c^Dlx5−6/y^*
*Kif5c*	2	LoF mutations (heterozygous)	Complex Cortical Dysplasia with Other Brain Malformations 2 (CDCBM2)	[22]	*Kif5c^+/−^; Kif5c^E237V^*
*Klhl15*	X	LoF mutations & deletions (hemizygous)	Intellectual disability	[28]	*Klhl15^−/y^*
*Larp7*	3	LoF mutations & duplications (homozygous)	Alazami syndrome	[3,35]	*Larp7^−/y^*
*Mbd5*	2	LoF mutations, translocation, duplications & deletions (heterozygous)	Autosomal Dominant Mental Retardation 1 (MRD1) & 2q23.1 duplication and deletion syndromes	[36,37,38]	*Mbd5^−/−^*
*Mecp2*	X	LoF mutations (heterozygous in females)	Rett syndrome, Atypical Rett Syndrome or Angelman-like Phenotype & Autism	[39,40]	*Mecp2^−/y^*
*Med12*	X	Missense mutations (hemizygous) p.(R961W) most frequent mutation	Lujan-Fryns syndrome, X-linked Ohdo syndrome & Opitz-Kaveggia syndrome	[41,42,43]	*Med12^−/y^; Med12^R961X^*
*Med17*	9	p.(L371P) missense mutation (homozygous)	Postnatal progressive microcephaly with seizures and brain atrophy	[44]	*Med17^L369P^*
*Med23*	10	p.(R617Q) missense mutation (homozygous)	Autosmal recessive intellectual disability 18	[45]	*Med23^−/−^; Med23^R617Q^*
*Med25*	7	Missense mutations (homozygous)	Autosmal recessive Charcot-Marie-Tooth type 2B2 disease & Basel-Vanagait-Smirin-Yosef syndrome (BVSYS)	[46,47]	*Med25^−/−^*
*mIR124a-2*	3	-	-	-	*Mir124^−/−^*
*mIR137*	3	Microdeletions (heterozygous)	Intellectual disability, autism & schizophrenia	[48,49]	*Mir137^−/−^*
*Nr1i3*	1	LoF mutations (heterozygous)	Core features of Kleefstra syndrome	[50]	*Nr1i3^−/−^*
*Pafah1b1*	11	LoF mutations, deletion & translocation (heterozygous)	Lissencephaly 1, Miller-Dieker lissencephaly syndrome & Subcortical laminar heterotopia	[51,52]	*Pafah1b1^−/−^*
*Phf6*	X	LoF Mutations (heterozygous in females & hemizygous in males)	Borjeson-Forssman-Lehmann syndrome (BFLS)	[53]	*Phf6^−/y^*
*Phf8*	X	LoF mutations & deletion (hemizygous)	Siderius X-linked Mental retardation syndrome (MRXSSD)	[54]	*Phf8^−/y^*
*Prps1*	X	LoF & GoF mutations (hemizygous)	LoF: Arts syndrome, X-linked recessive Charcot-Marie-Tooth disease 5 & X-linked non syndromic hearing loss (NSHL) vs. GoF: PRPS-related Gout syndrome & Phosphoribosylpyrophosphate Synthetase Superactivity	[55,56,57]	*Prps1* * ^−^ * * ^/y^ * *; Prps1^Camk2a/y^; Prps1^Cspg4/y^*
*Ptchd1*	X	LoF mutations & deletion (hemizygous)	X-linked Autism Susceptibility (AUTSX4)	[58,59]	*Ptchd1^−/y^*
*Scaper*	9	LoF mutation (homozygous & compound heterozgous)	Retinitis pigmentosa with intellectual disability	[60]	*Scaper^−/−^*
*Setbp1*	18	GoF mutations (Schinzel-Giedion Syndrome) & LoF mutations (autosmal dominant mental retardation 29) (heterozygous)	Schinzel-Giedion Syndrome vs. Autosomal dominant Mental Retardation syndrome 29 (MRD29)	[61,62,63]	*Setbp1^−/−^*
*Stard8*	X	-	-	-	*Stard8^−/y^*
*Taf2*	15	Missense mutations (homozygous)	Autosomal recessive Mental retardation 40 (MRT40)	[3,64]	*Taf2^−/−^*
*Tti2/C8orf41*	8	Missense mutations (homozygous)	Autosomal recessive Mental retardation 39 (MRT39)	[3,65]	*Tti2^−/−^*
*Tubb3*	8	LoF mutations (heterozygous)	Complex Cortical Dysplasia with Other Brain Malformations (CDCBM1) & Congenital Fibrosis of Extraocular Muscles 3a (CFEOM3A)	[66,67]	*Tubb3^M388V^*
*Tubg1*	11	Missense mutations (heterozygous)	Complex Cortical dysplasia with other brain malformations 4 (CDCBM4)	[22]	*Tubg1^−/−^; Tubg1^Y92C^*
*Wdr62*	7	LoF mutations (homozygous)	Autosomal recessive primary Microcephaly 2 with or without cortical malformations	[68,69]	*Wdr62^−/−^*
*Zc4h2*	X	Missense & in frame insertion (hemizygous in males), LoF mutations, splice site, missense & (partial) gene deletions (heterozygous in females)	ZC4H2-Associated Rare Disorders (ZARD), previously known as Wieacker-Wolff syndrome (WRWF)	[70,71]	*Zc4h2^−/y^*

## Data Availability

The data presented in this study are available upon request.

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
