# Peer review of "Large-Scale Functional Assessment of Genes Involved in Rare Diseases with Intellectual Disabilities Unravels Unique Developmental and Behaviour Profiles in Mouse Models"

_biomedicines, 2022, doi:10.3390/biomedicines10123148_

Round 1

Reviewer 1 Report

This is a very comprehensive phenotypic evaluation of mouse models of IDD.  These results of these analyses will be very helpful for investigators who are interested in studying animal models of these genes.  The paper is very well presented, the data are rigorous and transparent. This study presents only those results in male mice, so additional studies to analyze similar phenotypes in female mice would be helpful for the research community.

Author Response

We agree with the comments of reviewer 1, and we know about the limitation of study that we stated in the initial cover letter and in the manuscript: the results of this comprehensive study are from male mice. Nevertheless, as stated by reviewer 1, we are convinced that it will be helpful for the research community. We will surely benefit from additional studies to analyze similar phenotypes in female mice.

Author Response

We addressed all the comments raised by reviewer 2. First, we did a careful check of the spelling with a proof editing company. The introduction and conclusion were improved. In addition, several of the references are taken out from the present version, and an update reference added for the number of genes involved in ID.

1) Check spelling, spacing etc.

As indicated above, an effort was made by including propositions of a proof-editing service completed by a thorough lecture to avoid any potential spelling, spacing or error typing.

2) Introduction and conclusion can be improved.

Several corrections were made in the introduction and discussion conclusion part as indicated in the “track changes version”

3) Results of all figures need to be clearly and easily visible.

The figures legends were further developed to clearly point out the issued results. Additional information is added in the main text. In addition, a legend is added for table 1.

Question:

  1. Parallel to genes involvement, possible role of epigenetic in intellectual disabilities?

We agree with the reviewer. The epigenetic is also involved in the ID disorders. And a sentence is added in the introduction “the proteins encoded by these genes act in various biological processes, such as transcription regulation, epigenetic modification and synaptic transmission.” It is the case for example for CDKL5 which, in addition to its proper involvement in CDKL5 Deficiency Disorder, is involved in the regulation of MECP2

 Activity. In addition, patients with pathogenic variants of one or the other gene present with overlapping clinical features.    

1 and 3. Why so many gene investigated in this study? Why the author interested to explore both known and unknown genes related to growth hormone and short stature?

This report is the result of a large-scale study performed in the frame of a multidisciplinary European funded program “GENCODYS” designed to gain pathways-based insights into mechanisms leading to cognitive dysfunction in humans. This research program exploited the strongly complementary expertise of groups renowned in each of their research fields, from clinicians to fundamental neuroscience experts. Based on clinician discoveries of genes involved in intellectual disabilities, particularly De novo genes, this effort is the first one, according to our knowledge, using systematic approach to generate and characterize appropriate mutant mouse lines for several of these genes, in a standardized way.